# Identification of Three Circulating MicroRNAs in Plasma as Clinical Biomarkers for Breast Cancer Detection

**DOI:** 10.3390/jcm12010322

**Published:** 2022-12-31

**Authors:** Shuang Wang, Lijuan Li, Mengmeng Yang, Xiaoyan Wang, Huan Zhang, Nan Wu, Kaichao Jia, Junchao Wang, Menghui Li, Lijuan Wei, Juntian Liu

**Affiliations:** 1Department of Cancer Prevention Center, Tianjin Medical University Cancer Institute and Hospital, Huanhu Xi Road, Hexi District, Tianjin 300060, China; 2National Clinical Research Center for Cancer, Tianjin’s Clinical Research Center for Cancer, Huanhu Xi Road, Hexi District, Tianjin 300060, China; 3Key Laboratory of Breast Cancer Prevention and Therapy, Tianjin Medical University, Ministry of Education, Huanhu Xi Road, Hexi District, Tianjin 300060, China; 4Institute of Radiation Medicine, Chinese Academy of Medical Sciences & Peking Union Medical College, Tianjin 300192, China; 5Department of Clinical Laboratory, Tianjin Medical University Cancer Institute and Hospital, Huanhu Xi Road, Hexi District, Tianjin 300060, China

**Keywords:** breast cancer, miRNAs, biomarkers, diagnosis, plasma, datasets, biological functions

## Abstract

The diagnostic value of microRNAs (miRNAs) for breast cancer (BC) is largely unknown. Here, our research aim was to explore new circulating miRNAs for BC diagnosis. First, we identified 14 common differentially expressed miRNAs in tissues by TCGA_BRCA and GSE97811 datasets and preliminarily validated them in serum by the GSE73002 dataset. Furthermore, we examined three plasma miRNAs in BC patients (n = 108) and healthy subjects (n = 103) by RT–PCR, namely, hsa-miR-100-5p, hsa-miR-191-5p and hsa-miR-342-3p. The levels of these three miRNAs in BC patients were higher than those in healthy controls (*p* < 0.05). The ROC curve analysis revealed that these three miRNAs had high diagnostic efficacy for BC and early-stage BC. The combination of hsa-miR-100-5p and hsa-miR-191-5p was the optimal combination for the diagnosis of BC and early-stage BC. Additionally, hsa-miR-100-5p was correlated with stage I–II, T1 stage, N0 stage and Luminal A subtype (*p* < 0.05). Hsa-miR-191-5p and hsa-miR-342-3p were irrelevant to TNM stage, T stage, N stage and molecular subtypes. Meanwhile, the biological function analysis indicated that these three miRNAs are mainly involved in the calcium signaling pathway, MAPK signaling pathway and microRNAs in cancer. In conclusion, these three miRNAs demonstrate a positive effect on detection and discovery in BC.

## 1. Introduction

Breast cancer (BC) has attracted worldwide attention as a crucial health concern among women, and it is considered a leading cause of female mortality. Although mortality rates in developed countries are decreasing, mortality trends in low- and middle-income countries are still increasing [1]. Among countries, China has the highest number of BC cases and deaths, and it mostly occurs in middle-aged women [2]. Therefore, effective screening and prevention strategies are the most beneficial measures to reduce BC incidence and mortality. For BC screening, mammography, breast ultrasonography and clinical breast examination have greatly improved the detection rate of BC [3]. 

Liquid biopsy is an effective noninvasive examination method in diagnosis and monitoring of tumors, which contains a variety of tumor-derived materials, such as protein, circulating DNA, circulating RNA, circulating tumor cells, extracellular vesicles, etc. However, the clinical value of commonly used liquid biopsy tumor biomarkers, including carbohydrate antigen 153 (CA153) and carcinoembryonic antigen (CEA), is better reflected in the dynamic monitoring of therapeutic effects in patients with BC rather than screening and diagnosis of BC [4,5]. In the field of liquid biopsy, novel circulating biomarkers require further discovery, which is a great challenge for BC diagnosis. MicroRNA (miRNA) is a type of noncoding RNA with a length of 19–25 nucleotides that binds to the 3′-end untranslated region of the target mRNA through base complementation and regulates target gene expression at the posttranscriptional level. Generally, the mature miRNA binds to specific regions of the mRNA to prevent mRNA translation or promote mRNA degradation [6,7]. The regulation of mature miRNA levels is critical for the development and differentiation of the disease. In the case of BC, existing studies have demonstrated that miRNA expression impacts occurrence, proliferation, invasion and metastasis [8,9,10]. Specific miRNAs in tissues, blood and body fluids, as high-quality tumor biomarkers, have assisted early diagnosis and predicted the poor prognosis of BC [11,12,13,14]. As a noninvasive test, an increasing number of circulating miRNAs have been reported to be stably present in the blood with high specificity and sensitivity for the diagnosis of BC [15].

In recent years, the application of sophisticated next-generation sequencing technology has made genetic research more widespread with the opening of public datasets with high-throughput sequencing weighted convenience for the analysis of disease gene signatures. The Cancer Genome Atlas (TCGA) dataset and Gene Expression Omnibus (GEO) datasets have established comprehensive miRNA expression information and clinical data, which could assist BC research in obtaining reliable data. Based on this, we analysed the information of meaningful miRNAs in public datasets and detected circulating microRNA expression levels in the plasma of healthy subjects and BC patients, intending to explore novel tumor biomarkers that are valuable for BC screening and diagnosis. 

## 2. Results

### 2.1. Study Workflow

In this study, different expressed miRNAs with the same trend were identified from TCGA_BRCA dataset and GSE97811 dataset. In the verification stage, serum samples from GSE73002 dataset were preliminarily verified and three interesting miRNAs were screened out, namely, hsa-miR-100-5p, hsa-miR-191-5p and hsa-miR-342-3p. Further, a case-control study was conducted to analyse the differential expression levels of these three miRNAs in BC patients and healthy controls, and to explore their relationship with clinical characteristics. Finally, the biological functions and target genes involved in these three miRNAs were obtained by biological function analysis. The research idea is shown in Figure 1.

### 2.2. Identification of Potential MiRNAs from Public Datasets

To screen significant potential circulating miRNAs in BC, we analysed three public datasets of microarray data. First, differential miRNAs between adjacent normal and BC tissues were identified. A total of 1102 BC tissues and 119 adjacent normal tissues were obtained from two different datasets (TCGA_BRCA and GSE97811). We identified differentially expressed miRNAs in tissue using |log fold change (FC)| > 1 and false discovery rate (FDR) < 0.05 as screening criteria to receive the differential results. There were 304 miRNAs (downregulated: 91; upregulated: 213) that were differentially and significantly expressed in TCGA_BRCA dataset (Figure 2A). In the GSE97811 dataset, the results revealed that 40 miRNAs (downregulated: 11; upregulated: 29) were differentially and significantly expressed (Figure 2B). By screening and verifying the same expression trend of miRNAs, we recognized 14 common genes, which included hsa-miR-145-5p, hsa-miR-99a-5p, hsa-miR-125b-5p, hsa-miR-100-5p, hsa-miR-205-5p, hsa-miR-195-5p, hsa-miR-93-5p, hsa-miR-191-5p, hsa-miR-106-5p, hsa-miR-342-3p, hsa-miR-200b-3p, hsa-miR-21-5p, hsa-miR-200a-3p and hsa-miR-141-3p. The differential expression of 14 miRNAs was plotted in a heatmap (Figure 2C,D) and is listed in Table 1.

Furthermore, the corresponding 14 serum miRNAs were preliminarily validated in the GSE73002 dataset, with 1280 BC patients and 2682 healthy controls. Notably, the results demonstrated that all 14 miRNAs were upregulated in BC patients (Table 2). From this, the expression trend of miRNA seems to be different in tissues and blood. Finally, three candidate miRNAs of interest and less studied were preferentially verified in plasma, namely, hsa-miR-100-5p, hsa-miR-191-5p and hsa-miR-342-3p.

### 2.3. Baseline Characteristics of BC Patients and Healthy Controls in the Validation Cohort

Plasma samples from 108 BC patients and 103 healthy controls were collected and tested for this study. The demographics and clinical characteristics of BC patients and healthy controls are listed in Table 3. There was no significant difference in age distribution between the two groups. After each sample was normalized with an internal reference and the relative expression levels were calculated, the results showed that the relative expression levels of hsa-miR-191-5p, hsa-miR-342-3p and hsa-miR-100-5p in the plasma of BC patients were significantly different compared with healthy controls (all *p* < 0.05). Interestingly, these results were consistent with the analytical results in the GSE73002 datasets. Meanwhile, there was no difference in the concentrations of CEA and CA153.

### 2.4. Evaluation of the Diagnostic Efficacy of MiRNA Biomarkers in the Validation Cohort

To assess whether these three miRNAs could be used to diagnose BC, ROC analysis was performed using the relative expression values of miRNAs. Compared with the healthy controls (hsa-miR-100-5p, median: 0.89, 95% CI: 1.13–1.66; hsa-miR-191-5p, median: 1.09, 95% CI: 0.90–2.53; hsa-miR-342-3p, median: 1.07, 95% CI: 1.03–1.35), there were significantly increased expression levels of hsa-miR-100-5p, hsa-miR-191-5p and hsa-miR-342-3p in the plasma of BC patients (hsa-miR-100-5p, median: 1.34, 95% CI: 1.49–2.01; hsa-miR-191-5p, median: 11.49, 95% CI: 13.20–18.38; hsa-miR-342-3p, median: 2.81, 95% CI: 2.84–3.63) (Figure 3A). The plasma levels of these three miRNAs were higher in BC patients with stage I–II disease (hsa-miR-100-5p, median: 1.35, 95% CI: 1.43–1.98; hsa-miR-191-5p, median: 11.49, 95% CI: 12.79–19.21; hsa-miR-342-3p, median: 2.62, 95% CI: 2.78–3.80) than in healthy subjects (Figure 4A).

Furthermore, we evaluated the values of these three miRNAs for the diagnosis of BC (Figure 3B) and early-stage BC (Figure 3C). The results of ROC analysis indicated that miRNA alone and the combined signature were competent for discriminating between BC patients and healthy controls. Among them, hsa-miR-191-5p has excellent diagnostic efficacy (diagnosis of BC, 0.9549 AUC, 86.11% sensitivity, 97.09% specificity; diagnosis of early-stage BC, 0.9551 AUC, 84.62% sensitivity, 97.09% specificity). Depending on the results of different allocation groups, hsa-miR-191-5p combined with hsa-miR-100-5p is the optimal combination for the diagnosis of BC and early BC screening (diagnosis of BC, 0.9615 AUC, 93.52% sensitivity, 93.20% specificity; diagnosis of early-stage BC, 0.9556 AUC, 93.59% sensitivity, 91.26% specificity). However, compared with these three miRNAs, CEA and CA153 have lower diagnostic efficacy (diagnosis of BC, 0.5679 AUC, 50.00% sensitivity, 66.02% specificity; diagnosis of early-stage BC, 0.5871 AUC, 55.14% sensitivity, 66.02% specificity). Each value of AUC, sensitivity, specificity and Youden index is shown in Table 4. The diagnostic evaluation of CEA and CA153 is shown in Table A1.

### 2.5. Correlation with Clinicopathology Features

Subsequently, the relationship between miRNA expression levels and clinicopathological features was connected. These results revealed that the TNM stage, tumor size (T1/T2–3) and regional lymph node involvement (N0/N1–3) of hsa-miR-191-5p and hsa-miR-342-3p had remarkable differences when compared with healthy controls (all *p* < 0.001) (Figure 4A–C). Nevertheless, it was independent of tumor size and number of lymph node metastases. In addition, the expression level of hsa-miR-100-5p was obviously increased in stage I–II (stage I–II, *p* = 0.0039; stage III, *p* = 0.1122), T1 stage (T1, *p* = 0.002; T2–3, *p* = 0.0769) and N0 stage (N0, *p* < 0.001; N1–3, *p* = 0.481). Interestingly, the plasma level of hsa-miR-100-5p was elevated in the Luminal A subtype but not the Luminal B, HER2 and TNBC subtypes (Luminal A, *p* = 0.0078; Luminal B, *p* = 0.0507; HER2, *p* = 0.2359; TNBC, *p* = 0.0975), and the plasma levels of hsa-miR-191-5p and hsa-miR-342-3p were highly expressed in all three subtypes of BC patients (*p* < 0.05) (Figure 4D). These results suggest that high levels of hsa-miR-191-5p and hsa-miR-342-3p might be influencing factors of BC and that high levels of hsa-miR-100-5p are closely related to Luminal A and early-stage BC.

### 2.6. Functional Enrichment for miRNA Target Genes

To explore the function of miRNAs, the target genes of these three miRNAs were identified by filtering the miRDB, miRTarBase and TargetScan datasets on target genes that appeared in more than two datasets (Figure 5A). Next, we forecasted their target mRNAs using the TCGA_BRCA dataset. Differentially expressed mRNAs were identified in 694 BC tissues and 66 adjacent normal tissues (Figure 5B). In total, 4998 mRNAs were differentially expressed, of which 1828 were downregulated and 2846 were upregulated. According to the principle of miRNAs negatively regulating mRNA, the network showed that these three miRNAs regulated multiple target genes (Figure 5C). Furthermore, analysis of the KEGG signaling pathway demonstrated that the target genes of the three miRNAs mainly participated in the calcium signaling pathway, mitogen-activated protein kinase (MAPK) signaling pathway and microRNAs in cancer (Figure 5D). The related genes are listed in Table A2.

## 3. Discussion

The detection of noninvasive tumor markers is a convenient way to discover cancer and a reliable clinical auxiliary diagnostic tool to improve the level of diagnosis and treatment monitoring. Many studies have focused on miRNAs that can stabilize their presence in the blood. Nonetheless, studies on circulating miRNAs as noninvasive markers are still in the developmental and immature stage. Therefore, our study filtered BC-related public datasets to obtain miRNAs with diagnostic capability and validated them in plasma.

Initially, we focused on two BC tissue datasets (TCGA_BRCA and GSE97811) and one BC serum dataset (GSE73002). Generally, the population in the TCGA dataset is different from the Chinese population, mostly from European and American countries. To explore candidate genes more suitable for the diagnosis of BC in the Chinese population, we filtered the GEO dataset and obtained two miRNA datasets of Asian populations from Japan: the GSE97811 and GSE73002 datasets. Based on these, we screened and analysed 14 common miRNAs that were differentially expressed in BC tissues and adjacent normal tissues (Figure 2C,D). Moreover, the differential expression of these miRNAs was identified in the serum of BC patients and healthy controls. Intriguingly, all miRNA levels were upregulated in the serum of BC patients (Table 2). Existing research revealed that hsa-miR-21-5p [16,17], hsa-miR-145-5p [18,19], hsa-miR-99a-5p [20,21,22], hsa-miR-125b-5p [17,23], hsa-miR-205-5p [24,25], hsa-miR-195-5p [26,27], hsa-miR-106b-5p [28,29] and hsa-miR-191-5p [17,30] have been proposed for the diagnosis of BC patients in tissues and blood. Likewise, hsa-miR-100-5p and hsa-miR-141-3p have been proven to be useful in the diagnosis of BC in tissues [31,32]. Maria Amorim et al. demonstrated that hsa-miR-200b-3p was highly expressed in luminal BC tissues and displayed independent prognostic ability for disease recurrence in luminal BC patients after endocrine therapy [33]. The level of hsa-miR-93-5p was highly expressed in exosomes of BC patients and associated with tumor recurrence and distant organ metastasis [34]. Circulating hsa-miR-342-3p is highly expressed in TNBC plasma but not in non-TNBC patients [35]. The differentially expressed miRNA of miR-200a-3p was upregulated in BC patients with sentinel lymph node metastasis [36]. In this context, we selected three miRNAs of interest and less studied for further validation, namely, hsa-miR-100-5p, hsa-miR-191-5p and hsa-miR-342-3p.

Previous studies have shown that the expression of hsa-miR-100-5p in BC tissues is lower than that in noncancer tissues, which participates in the differentiation of breast cancer stem cells and inhibits the proliferation and invasion of BC [37,38]. For neoadjuvant chemotherapy, upregulated hsa-miR-100-5p was associated with better prognosis in BC patients [31]. Higher expression of plasma hsa-miR-100-5p was present in BC patients with hormone receptor positivity and metastasis after dovitinib treatment, and the circulating hsa-miRNA-100-5p levels were able to separate patients with resistant disease from sensitive patients [31]. Here, we verified the diagnostic value of circulating hsa-miR-100-5p for BC for the first time. We proposed that plasma hsa-miR-100-5p in BC patients was significantly higher than that in healthy controls (Figure 3A), especially in early-stage BC patients (Figure 4A), with an AUC value of 0.6254 (Table 4). Additionally, this study also confirmed that there was no significant correlation between the expression level of hsa-miR-100-5p and T2–3 stage and N1–3 stage, suggesting that a higher level of hsa-miR-100-5p had a better prognosis in BC patients. In addition, the higher expression of miR-100-5p mainly occurred in the Luminal A subtype of BC rather than the Luminal B, HER2 and TNBC subtypes (Figure 4D). Conversely, Annalisa Petrelli et al. indicated that the downregulated hsa-miR-100-5p was assumed to be in the luminal subtype of BC patient tissue and served as a predictor of endocrine responsiveness and prognosis in patients with operable luminal BC, and as a novel biomarker in patients with luminal BC [39]. In brief, the opposite difference levels of hsa-miRNA-100-5p in tissues and plasma suggested its close relationship with the development of luminal BC.

Among these 14 candidate genes, the differentially expressed hsa-miR-191-5p was identified to have the largest fold change in the serum dataset of GSE73002 (Table 2). Some studies have identified that hsa-miR-191-5p, which is highly expressed in BC tissues, is associated with lymph node metastasis [17]. Additionally, the higher expression of circulating hsa-miR-191-5p was confirmed in BC patients in small populations from Kazakhstan and Mexico [30]. Nonetheless, we extremely preferred realizing the hsa-miR-191-5p signature in Chinese plasma. Notably, our results demonstrated that hsa-miR-191-5p was significantly overexpressed in the plasma of BC patients, and the AUC value was 0.9549, with 86.11% sensitivity and 97.09% specificity. When applied in the early diagnosis of BC, the value of hsa-miRNA-191-5p was 0.9551, with 84.62% sensitivity and 97.09% specificity. Indisputably, hsa-miR-191-5p is an excellent noninvasive biomarker for BC detection, but the current research results of TNM stage, tumor size, lymph node metastasis and molecular subtypes suggest that it is independent of prognosis.

Recently, the action of hsa-miR-342-3p has been confirmed to be involved in BC metastasis through exosomes or extracellular vesicles [40,41]. In a small study of inflammatory breast cancer (IBC) and non-IBC, hsa-miR-342-3p was proposed to be significantly upregulated in non-IBC compared with IBC, but not compared with healthy subjects [42]. For circulating biomarker research, our results revealed that BC patients had higher expression of hsa-miR-342-3p than healthy controls, and this expression was not connected with TNM stage, tumor size or lymph node metastasis. Moreover, ROC analysis for BC diagnosis showed that the AUC was 0.8969, with 75.93% sensitivity and 92.23% specificity. Ultimately, ROC analysis for the early diagnosis of BC showed that the AUC was 0.8950, with 89.74% sensitivity and 76.70% specificity (Table 4). Visibly, hsa-miR-342-3p is also a high-quality biomarker for BC detection. In contrast, V Y Shin et al. [35] proposed that a high level of hsa-miR-342-3p was expressed in TNBC rather than non-TNBC, whereas this study indicated that hsa-miR-342-3p was undifferentiated in plasma between TNBC and non-TNBC patients. The small sample size might have influenced the result.

Combined diagnosis of multiple indicators can improve the specificity and sensitivity of a single marker. Relying on the pairing between these three miRNAs, the optimal combination for BC diagnosis and early detection was hsa-miR-191-5p combined with hsa-miR-100-5p. For the diagnosis of BC, the AUC value of this combination is 0.9615, with 93.52% sensitivity and 93.20% specificity. For the early diagnosis of BC, the AUC value of this combination is 0.9556, with 92.31% sensitivity and 93.20% specificity. However, the AUC value of CEA combined with CA153 for BC was 0.5679, with 50.00% sensitivity and 66.02% specificity. For the diagnosis of early-stage BC, the AUC value was 0.5871, with 55.15% sensitivity and 66.02% specificity. Obviously, the combination of hsa-miR-191-5p and hsa-miR-100-5p is superior to the traditional tumor markers in the detection of BC.

MiRNAs not only play a crucial role in diagnosis but also contribute to biological functions in BC. Here, we still analysed the target gene pathways of hsa-miR-100-5p, hsa-miR-191-5p and hsa-miR-342-3p. According to the relationship between miRNAs negatively regulating mRNAs, we recognized differentially expressed mRNAs by using mRNA microarray data in the TCGA_BRCA dataset and then predicted the related signaling pathways of target genes. Our analysis verified that these three miRNA signature target genes were mainly involved in the calcium signaling pathway, MAPK signaling pathway and microRNAs in cancer (Figure 5D). A large number of studies have confirmed the correlation between BC and these three signaling pathways [43,44,45,46,47].

In particular, platelet-derived growth factor receptor alpha (PDGFRA) and fibroblast growth factor receptor 3 (FGFR3) appeared in all three pathways and were the target genes of hsa-miR-342-3p and hsa-miR-100-5p, respectively (Figure 5C). PDGFRA encodes PDGFRα, a member of the PDGF receptor family, which belongs to the family of receptor tyrosine kinases (RTKs) [48]. A previous study reported that hsa-miR-342-3p inhibited the proliferation, migration, invasion and G1/S phase transition of HTR8/SVneo cells by suppressing PDGFRA in women with preeclampsia [49]. Recently, some studies suggested a more frequent PDGFRA activation signature than non-IBC samples, and the PDGFRA activation signature is associated with shorter metastasis-free survival in IBC [50,51]. However, the targeting relationship between hsa-miR-342-3p and PDGFRA has not been explored in BC. Moreover, FGFR3 was significantly related to poor overall survival in BC and was confirmed to be a candidate therapeutic target in FGFR3-associated BC [52]. The regulatory connection between miR-100 and FGFR3 has been described in osteosarcoma [53], bladder cancer [54], glioblastoma [55], lung cancer [56], prostate cancer [57] and pancreatic cancer [58], but has never been reported in BC.

In addition, early growth response 1 (EGR1) and regulator of G-protein signaling 2 (RGS2), the target genes of hsa-miR-191-5p, are mainly involved in the gonadotropin-releasing hormone (GnRH) signaling pathway, apelin signaling pathway, oxytocin signaling pathway and cGMP-PKG signaling pathway (Table A2), which are interrelated to BC [59]. Gianpiero Di Leva et al. demonstrated that hsa-miR-191-5p protects ERα-positive BC cells from hormone starvation-induced apoptosis through the suppression of the tumor suppressor EGR1 [60]. Some studies have indicated that RGS2 has a tumor suppressor function and that low expression of RGS2 is associated with a significantly poorer overall survival rate [61,62]. However, whether hsa-miR-191-5p aggravates BC development by mediating RGS2 needs to be further explored in the future.

In conclusion, our results incorporated findings from public datasets and verified them in the plasma of BC patients and healthy individuals, which supports the diagnostic value of circulating hsa-miR-100-5p/ hsa-miR-191-5p/ hsa-miR-342-3p for detecting BC. In particular, these three miRNA signatures have prominent advantages compared with traditional tumor biomarkers and provide a new noninvasive means for clinical diagnosis of BC. Nevertheless, we need to further expand the sample size and verify its utility in multiple centers. In addition, our results lacked the verification of the expression levels of these three miRNAs in BC tissues, and could not reflect the difference in their diagnostic advantages in plasma and tissues. The detection of plasma miRNA has the characteristics of convenient sampling, high sensitivity and low detection cost. However, in terms of miRNA detection, more mature standardized detection methods and high-quality PCR assay platforms are needed to support and promote the conversion of miRNA as a marker of BC diagnosis to clinical practice. In the future, it will be necessary to evaluate the role of these three miRNAs in the occurrence of BC, and these studies will provide a basis for the pathogenesis and early diagnosis of BC.

## 4. Materials and Methods

### 4.1. Data Acquisition

Three public datasets were analysed in this study, including two tissue datasets (TCGA_BRCA and GSE97811) and one serum dataset (GSE73002). TCGA_BRCA data were downloaded from the BC project of TCGA datasets (https://portal.gdc.cancer.gov). GSE97811 and GSE73002 data were downloaded from GEO datasets (https://www.ncbi.nlm.nih.gov/gds). In TCGA_BRCA datasets, the clinical and miRNA sequences of 1057 BC tissues and 103 adjacent normal tissues were available for analysing differentially expressed miRNAs. Additionally, transcription data of 694 BC tissues and 66 adjacent normal tissues were collected to investigate differentially expressed mRNAs. BC data in TCGA were updated on 17 January 2021. In the GSE97811 dataset, the clinical data, miRNA-sequence data of 45 BC tissues and 16 adjacent normal tissues were used to analyse differentially expressed miRNAs. In the GSE73002 datasets, miRNA-sequence information of 1280 BC patients and 2682 healthy controls was used to analyse differentially expressed miRNAs in serum.

### 4.2. Subject Samples for Validation of miRNAs

All participants were recruited through Tianjin Medical University Cancer Institute and Hospital (Tianjin, China) from May 2022 to August 2022, including 108 female BC patients and 103 healthy controls. BC patients were newly admitted patients who had not been treated with radiation, chemotherapy or surgery. Healthy controls were determined to be free of BC and other malignancies and were recruited from the Department of Cancer Prevention Center. Tumor node-metastasis (TNM) stage and molecular subtypes of BC patients were based on international guidelines [63], including tumor size, regional lymph node involvement, and distant metastases, and the expression of estrogen receptor (ER), progesterone receptor (PR), human epidermal growth factor receptor 2 (HER2) and Ki-67. The study was conducted in accordance with the Declaration of Helsinki, and the protocol was approved by the Ethics Committee of Tianjin Medical University Cancer Institute and Hospital (bc2022142).

### 4.3. Sample Processing and miRNA Extraction

Blood samples were collected in EDTA-containing tubes and centrifuged at 3500 rpm for 10 min. The isolated plasma was stored at −20 °C until further use. MagZol LS reagent and HiPure Liquid RNA Mini Kit (Magen, Guangzhou, China) reagents were used according to the manufacturer’s recommended protocol. A 250 µL plasma sample was added to 0.75 mL TRIzol LS reagent. After separation of the aqueous phase, 1.5 volumes of 100% ethanol were added to the aqueous phase, and the mixture was loaded into a HiPure Viral Mini Column according to the manufacturer’s instructions. RNA samples were finally eluted into 28 µL RNase-free water and stored at −20 °C. The concentration of all miRNA samples was quantified by a NanoDrop 2000 spectrophotometer (Thermo Scientific, Wilmington, DE, USA).

The serum CEA and CA15-3 levels were detected using the Roche Cobas e601 automatic immunoassay analyser (Roche Diagnostics, Basel, Switzerland) and kits from the manufacturer.

### 4.4. MiRNA Validation by Quantitative Reverse Transcription (qRT–PCR)

MiRNAs were reverse transcribed into cDNA by using a reverse transcriptase of T100TM Thermal Cycler (Bio-Rad, California, America) with miRNA 1st Strand cDNA Synthesis Kit (by stem–loop) (Vazyme, Nanjing, China). The RT procedure was 25 °C for 5 min, 50 °C for 15 min, and then 85 °C for 5 min. PCR was performed using a LightCycler 480 Real-Time PCR System (Roche Diagnostics, Basel, Switzerland) with miRNA Universal SYBR qPCR Master Mix (Vazyme, Nanjing, China). The PCR procedure was 95 °C for 5 min, followed by 40 cycles of 95 °C for 10 s and 60 °C for 30 s, with a final extension at 95 °C for 15 s, 60 °C for 60 s and 95 °C for 15 s. The miRNA-specific primer sequences were designed based on the miRNA sequences obtained from the miRBase database (http://mirbase.org/), and the specific primer sequences are listed in Table 5. U6 was employed to normalize the expression of miRNAs. The relative expression of miRNA was calculated by the 2^−∆∆Cq^ method. ∆Cq represents the Cq value of each miRNA minus the Cq value of the corresponding internal reference. The mean ∆Cq value of miRNA in healthy individuals as a control, then the ∆Cq value of each miRNA, was subtracted from the mean ∆Cq value of healthy controls to obtain ∆∆Cq, and finally, the calculated 2^−∆∆Cq^ value (fold change of expression) was applied to diagnostic efficacy analysis and clinical characteristics correlation analysis.

### 4.5. Target Prediction Analysis and Functional Analysis

The target genes of miRNAs were merged with the miRDB (http://www.mirdb.org/), miRTarBase (http://mirtarbase.mbc.nctu.edu.tw/), and TargetScan (http://www.targetscan.org/) databases. All data were updated on 5 February 2021. The corresponding target mRNA was determined to be the target miRNA gene when present in at least two databases. Kyoto Encyclopedia of Genes and Genomes (KEGG) pathway analysis was performed to analyse the signaling pathways of the miRNA-related target genes.

### 4.6. Data and Statistical Analysis

Statistical calculations of TCGA_BRCA and GSE97811 datasets were analysed using R (version 3.6.3) (Auckland, New Zealand). The edgeR package was used to analyse differential miRNAs and mRNAs, using |log fold change (FC)| > 1 and false discovery rate (FDR) < 0.05 as the screening criteria. The VennDiagram package was used to plot the Venn diagram. ClusterProfiler and org.Hs.e.g.db were used to perform KEGG pathway analysis. Cytoscape 3.7.2 was utilized to map the relationship between miRNAs and target genes. The significance of plasma miRNA levels and microarray data of the GSE73002 dataset were determined by the Mann–Whitney test or T test. Receiver operator characteristic (ROC) curves were established for discriminating patients with or without BC. The optimal sensitivity and specificity from ROC curves were identified by a general approach. Statistical analysis was performed by Statistical Package for Social Sciences (SPSS) version 19 (Armonk, NY, USA) and GraphPad Prism 6 software (San Diego, CA, USA). All *p* values are two-sided, and *p* < 0.05 was considered statistically significant.

## Figures and Tables

**Figure 1 jcm-12-00322-f001:**
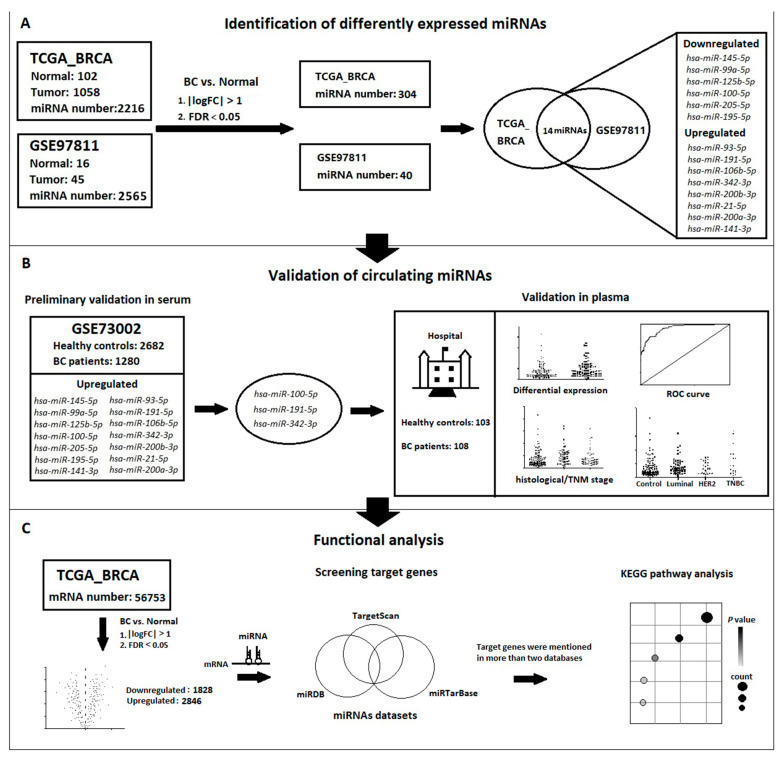
Workflow of this research. (**A**) Identification of differently expressed miRNAs in TCGA_BRCA and GSE97811 datasets. (**B**) Validation of circulating miRNAs in GSE73002 dataset and case-control study. (**C**) Biological function analysis of miRNA target genes.

**Figure 2 jcm-12-00322-f002:**
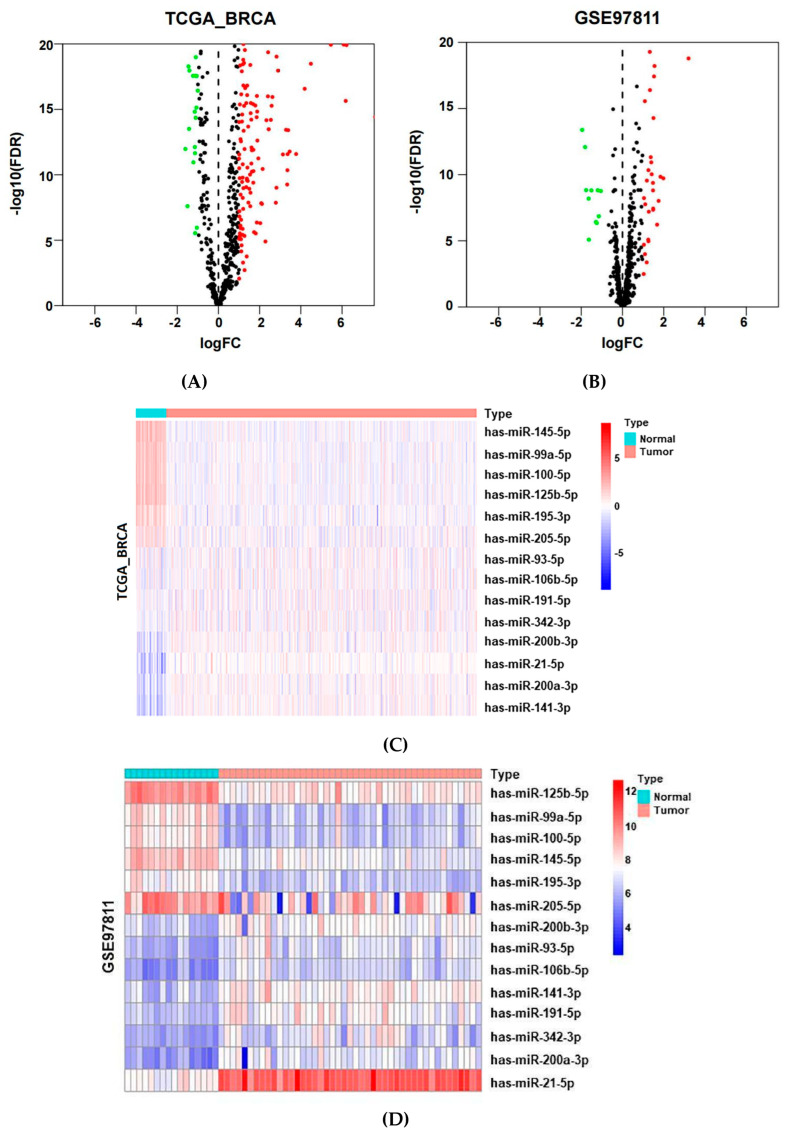
Identification of potential miRNAs from public datasets. (**A**) Volcano plot of differentially expressed miRNAs in the TCGA_BRCA dataset (downregulated: 91; upregulated: 213). (**B**) Volcano plot of differentially expressed miRNAs in the GSE97811 dataset (downregulated: 11; upregulated: 29). (**C**,**D**) Heatmap of 14 common miRNAs with the same expression trend in the TCGA_BRCA and GSE97811 datasets.

**Figure 3 jcm-12-00322-f003:**
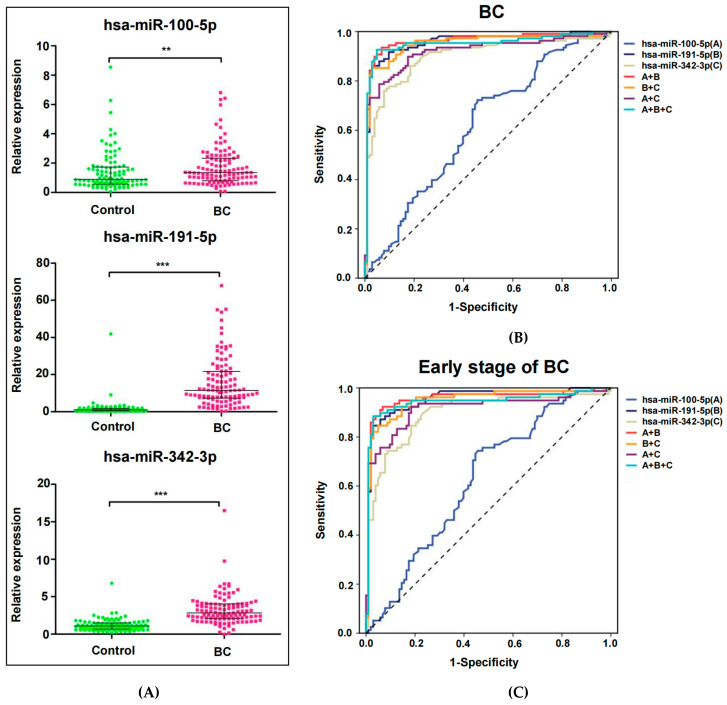
Evaluation of the diagnostic efficacy of hsa-miRNA-100-5p/hsa-miR-191-5p/hsa-miR-342-3p in the validation cohort. (**A**) Relative expression of these three miRNAs in the plasma of BC patients and healthy controls. (**B**) Receiver operating characteristic (ROC) curve analysis of these three miRNAs to discriminate BC patients from healthy controls. (**C**) ROC curve analysis of these three miRNAs to discriminate BC patients with early-stage disease from healthy controls. The scatter plot marks the interval between the 25th and 75th percentiles. Statistically significant differences were determined using Mann–Whitney tests. ** *p* < 0.01, *** *p* < 0.001.

**Figure 4 jcm-12-00322-f004:**
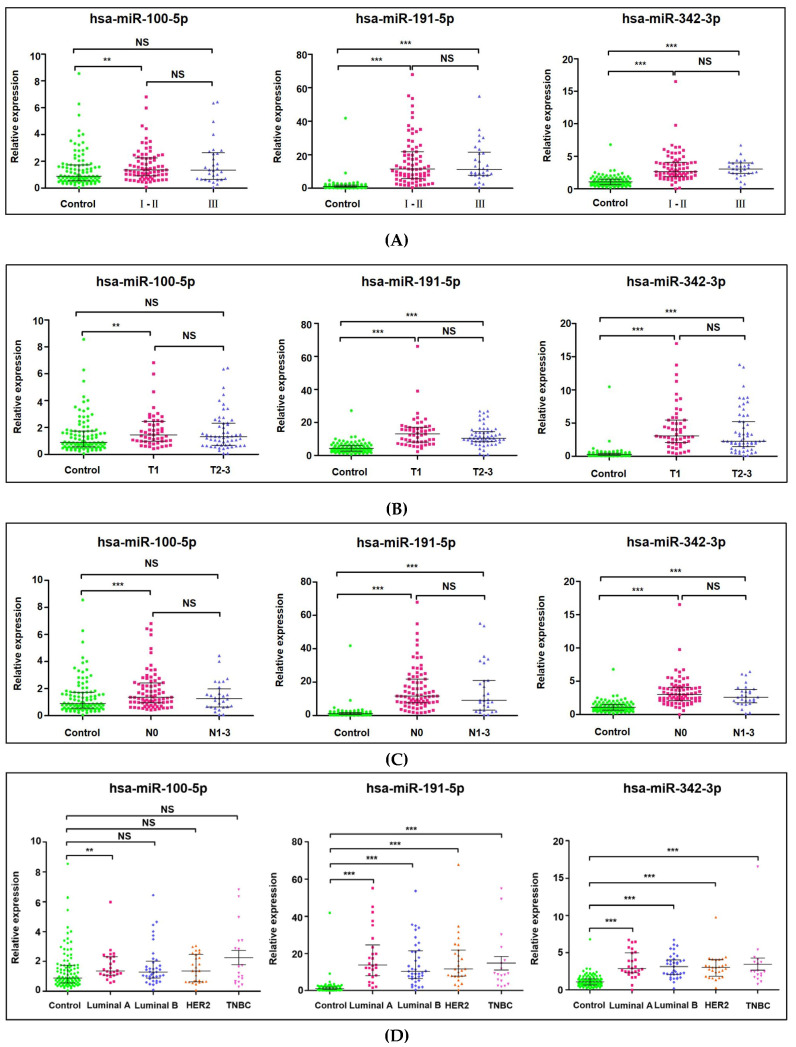
Correlation between the relative expression of hsa-miR-100-5p/hsa-miR-191-5p/hsa-miR-342-3p and clinicopathological features. (**A**) Relative expression of these three miRNAs in stage I–II and stage III (stage I–II: 78, stage III: 30); (**B**) Relative expression of these three miRNAs in T1 and T2–3 stage (T1 stage: 51, T2–3 stage: 57); (**C**) Relative expression of these three miRNAs in N0 and N1–3 stage (N0 stage: 80, N1–3 stage: 28); (**D**) Relative expression of these three miRNAs in Luminal A, Luminal B, HER2, and TNBC subtypes (Luminal A: 26, Luminal B: 37, HER2: 27, TNBC: 18). The lines inside the scatter plots represent the medians. The scatter plot marks the interval between the 25th and 75th percentiles. Statistically significant differences were determined using Mann–Whitney tests. NS: not significant. ** *p* < 0.01, *** *p* < 0.001.

**Figure 5 jcm-12-00322-f005:**
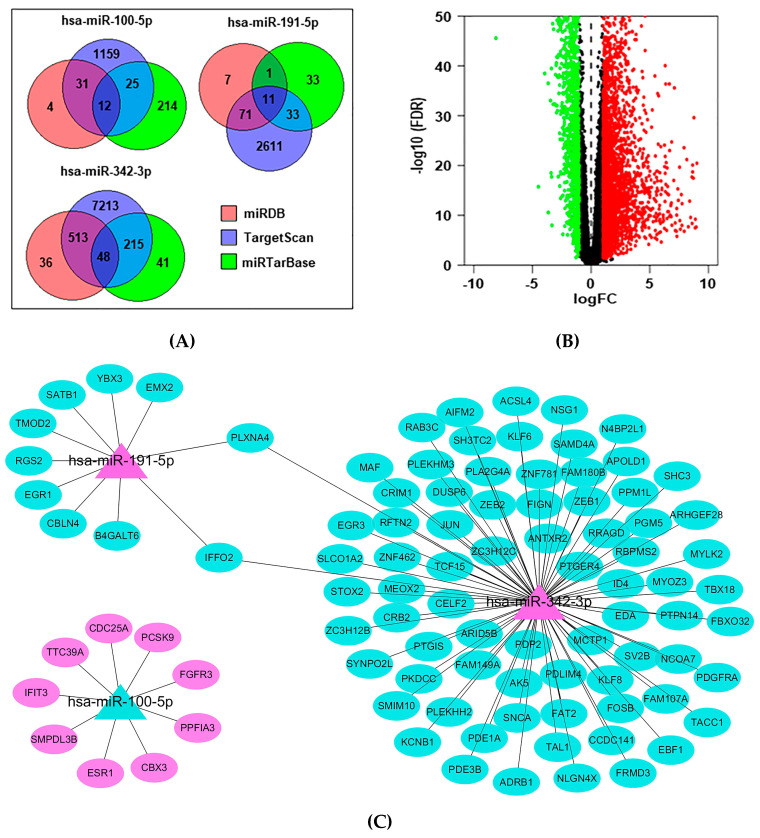
Target gene recognition and biological function exploration of hsa-miR-100-5p/hsa-miR-191-5p/hsa-miR-342-3p. (**A**) The Venn diagram of the target genes of these three miRNAs filtrated from miRDB, miRTarBase and TargetScan datasets, and the target genes appearing in more than two datasets were considered to be target genes of miRNA (hsa-miR-100-5p: 68, hsa-miR-191-5p: 115, hsa-miR-342-3p: 776) (**B**) Volcano plot of differentially expressed mRNAs in TCGA_BRCA dataset (Upregulated: 2846, Downregulated: 1828) (**C**) Target genes with differential expression of these three miRNAs were plotted in the network (hsa-miR-100-5p: 9, hsa-miR-191-5p: 10, hsa-miR-342-3p: 77) (**D**) The top 20 KEGG pathway analyses of these three miRNAs target genes are plotted in the bubble diagram.

**Table 1 jcm-12-00322-t001:** The expression of 14 common miRNAs in BC patients between TCGA_BRCA and GSE97811 datasets.

ID	TCGA_BRCA			GSE97811		
	logFC	*p*	FDR	logFC	*p*	FDR
hsa-miR-145-5p	−2.29	6.23 × 10^−215^	8.75 × 10^−213^	−1.51	1.00 × 10^−10^	1.55 × 10^−09^
hsa-miR-99a-5p	−1.96	1.19 × 10^−97^	4.41 × 10^−96^	−1.76	9.04 × 10^−11^	1.47 × 10^−09^
hsa-miR-125b-5p	−1.83	2.24 × 10^−118^	1.31 × 10^−116^	−1.81	1.98 × 10^−14^	8.25 × 10^−13^
hsa-miR-100-5p	−1.76	1.90 × 10^−85^	5.82 × 10^−84^	−1.64	4.96 × 10^−10^	6.48 × 10^−09^
hsa-miR-205-5p	−1.41	3.23 × 10^−19^	1.09 × 10^−18^	−1.63	1.26 × 10^−06^	8.13 × 10^−06^
hsa-miR-195-5p	−1.24	7.99 × 10^−44^	8.65 × 10^−43^	−1.29	4.28 × 10^−08^	3.75 × 10^−07^
hsa-miR-93-5p	1.26	6.90 × 10^−34^	5.11 × 10^−33^	1.26	1.78 × 10^−12^	4.56 × 10^−11^
hsa-miR-191-5p	1.32	3.36 × 10^−36^	2.78 × 10^−35^	1.49	4.25 × 10^−09^	4.49 × 10^−08^
hsa-miR-106b-5p	1.34	6.83 × 10^−49^	8.89 × 10^−48^	1.37	1.58 × 10^−13^	4.77 × 10^−12^
hsa-miR-342-3p	1.68	4.42 × 10^−34^	3.34 × 10^−33^	1.68	7.28 × 10^−08^	5.99 × 10^−07^
hsa-miR-200b-3p	1.83	1.20 × 10^−48^	1.54 × 10^−47^	1.27	6.15 × 10^−09^	6.30 × 10^−08^
hsa-miR-21-5p	2.28	1.70 × 10^−126^	1.20 × 10^−124^	3.20	7.37 × 10^−22^	1.64 × 10^−19^
hsa-miR-200a-3p	2.35	1.84 × 10^−63^	3.81 × 10^−62^	1.84	6.24 × 10^−12^	1.43 × 10^−10^
hsa-miR-141-3p	2.7	5.05 × 10^−83^	1.48 × 10^−81^	1.48	1.02 × 10^−10^	1.55 × 10^−09^

FC: fold change; FDR: false discovery rate

**Table 2 jcm-12-00322-t002:** The quantile normalized values of healthy controls and BC patients in the GSE73002 dataset.

ID	Healthy Controls (mean ± SD)	Breast Cancer Patients (mean ± SD)	*p*
hsa-miR-145-5p	2.12 ± 1.07	4.78 ± 1.19	<0.001
hsa-miR-99a-5p	2.01 ± 1.07	4.75 ± 1.32	<0.001
hsa-miR-125b-5p	2.28 ± 0.83	4.49 ± 1.16	<0.001
hsa-miR-100-5p	1.97 ± 1.08	4.60 ± 1.33	<0.001
hsa-miR-205-5p	2.40 ± 0.95	4.68 ± 1.14	<0.001
hsa-miR-195-5p	1.92 ± 1.02	4.35 ± 1.19	<0.001
hsa-miR-93-5p	2.29 ± 1.12	4.82 ± 1.11	<0.001
hsa-miR-191-5p	2.37 ± 1.23	6.20 ± 1.38	<0.001
hsa-miR-106b-5p	2.06 ± 1.08	4.60 ± 1.10	<0.001
hsa-miR-342-3p	2.86 ± 0.70	4.79 ± 1.03	<0.001
hsa-miR-200b-3p	2.02 ± 1.02	4.53 ± 1.27	<0.001
hsa-miR-21-5p	2.07 ± 1.14	4.85 ± 1.20	<0.001
hsa-miR-200a-3p	1.89 ± 1.03	4.43 ± 1.28	<0.001
hsa-miR-141-3p	1.81 ± 1.02	4.24 ± 1.25	<0.001

**Table 3 jcm-12-00322-t003:** The demographics and clinical characteristics of BC patients and healthy controls in the validation cohort.

Variable	Breast Cancer	Healthy Control	*p*
Number	108	103	
Age (mean ± SD)	52.79 ± 9.55	50.01 ± 13.49	0.087
Molecular subtype, N (%)			
Luminal A	26 (24.07)	NA	
Luminal B	37 (34.26)	NA	
HER2	27 (25.00)	NA	
TNBC	18 (16.67)	NA	
Stage, N (%)			
I	9 (8.33)	NA	
II	69 (63.89)	NA	
III	30 (27.78)	NA	
T, N (%)			
T1	51 (47.22)	NA	
T2	53 (49.07)	NA	
T3	4 (3.70)	NA	
N, N (%)			
N0	80 (74.07)	NA	
N1	21 (19.44)	NA	
N2	4 (3.70)	NA	
N3	3 (2.78)	NA	
CEA (μg/L) (median, range)	1.24 (0.92, 2.05)	1.17 (0.82, 1.75)	0.296
CA153 (U/mL) (median, range)	8.40 (5.63, 12.27)	9.36 (7.52, 12.10)	0.076
2^−ΔΔCq^ of hsa-miR-100-5p	1.34 (0.79, 2.33)	0.89 (0.55, 1.73)	0.0033
(median, range)			
2^−ΔΔCq^ of hsa-miR-191-5p	11.49 (7.35, 21.61)	1.09 (0.63, 1.81)	<0.001
(median, range)			
2^−ΔΔCq^ of hsa-miR-342-3p	2.81 (2.07, 4.04)	1.07 (0.65, 1.48)	<0.001
(median, range)			

T: tumor size; N: node; CEA: carcinoembryonic antigen; CA153:carbohydrate antigen 153; NA: not available; 2^−ΔΔCq^: relative expression of miRNA.

**Table 4 jcm-12-00322-t004:** Diagnostic efficiency evaluation indicators of hsa-miRNA-100-5p, hsa-miRNA-191-5p and hsa-miRNA-342-3p.

ID	BC					Early Stage of BC				
	AUC	Sensitivity	Specificity	Youden Index	*p*	AUC	Sensitivity	Specificity	Youden Index	*p*
	(95%CI)	(%)	(%)			(95%CI)	(%)	(%)		
hsa-miR-100-5p	0.6171	72.22	54.37	0.2659	0.003	0.6254	74.36	54.37	0.2873	0.004
(A)	(0.5411–0.6932)					(0.5444–0.7064)				
hsa-miR-191-5p	0.9549	86.11	97.09	0.8320	<0.001	0.9551	84.62	97.09	0.8170	<0.001
(B)	(0.9246–0.9852)					(0.9235–0.9868)				
hsa-miR-342-3p	0.8969	75.93	92.23	0.6816	<0.001	0.8950	89.74	76.70	0.6644	<0.001
(C)	(0.8510–0.9428)					(0.8440–0.9460)				
(A + B) *	0.9615	93.52	93.20	0.8672	<0.001	0.9556	92.31	93.20	0.8551	<0.001
	(0.9326–0.9905)					(0.9193–0.9919)				
(B + C) *	0.949	85.19	97.09	0.8227	<0.001	0.9482	84.62	95.15	0.7976	<0.001
	(0.9157–0.9823)					(0.9124–0.9840)				
(A + C) *	0.9187	78.70	94.17	0.7288	<0.001	0.9142	92.31	81.55	0.7386	<0.001
	(0.8781–0.9594)					(0.8660–0.9624)				
(A + B + C) *	0.9504	92.59	95.15	0.8774	<0.001	0.9431	88.46	97.09	0.8555	<0.001
	(0.9162–0.9845)					(0.9012–0.9850)				

* Combined diagnosis of three miRNAs paired with each other.

**Table 5 jcm-12-00322-t005:** Primer sequences of candidate miRNAs.

Name	Primer Sequence
hsa-miR-100-5p stem-loop primer	GTCGTATCCAGTGCAGGGTCCGAGGTATTCGCACTGGATACGACCACAAG
hsa-miR-100-5p forward primer	GCGAACCCGTAGATCCGAA
hsa-miR-191-5p stem-loop primer	GTCGTATCCAGTGCAGGGTCCGAGGTATTCGCACTGGATACGACCAGCTG
hsa-miR-191-5p forward primer	CGCAACGGAATCCCAAAAG
hsa-miR-342-3p stem-loop primer	GTCGTATCCAGTGCAGGGTCCGAGGTATTCGCACTGGATACGACACGGGT
hsa-miR-342-3p forward primer	GCGTCTCACACAGAAATCGC
U6 forward primer	CTCGCTTCGGCAGCACA
U6 reverse primer	AACGCTTCACGAATTTGCGT

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
