# Peer review of "Identification of Three Circulating MicroRNAs in Plasma as Clinical Biomarkers for Breast Cancer Detection"

_jcm, 2022, doi:10.3390/jcm12010322_

Round 1

Reviewer 1 Report

Dear Authors,

The work presented has a very interesting proposal, with an impact on clinical practice. However, two main points caught my attention, which caused the text to lose its impact:

 1) The authors set out to validate the diagnostic potential of circulating miRNA in patients with breast cancer. In the Liquid Biopsy era, this is a very relevant, current and original topic. However, at no time was the term “liquid biopsy” mentioned, which distracts the reader's attention.

2) Results: Although the expression data have been mentioned as relative, they are presented in the form of Delta Cq, which is nothing more than the normalization of the initial result. As it stands, it misleads readers who are less prepared or less accustomed to the qPCR technique, since the higher the Cq delta, the lower the expression of a gene.

3) In material and methods, the authors cite the comparison by the classic formula of 2^DDCq, but we do not have the graphical results following this logic: apparently, each sample was plotted with its value of DCq, without making a relative comparison (there is no calibrator with value 1 and fold change described). In the text, there is no mention of the proportionality of the difference in expression, which leads me to believe that there was no relationship between the results. Thus, I believe that the entire description of the results and discussion would be compromised.

I strongly suggest that the presentation of the results be reformulated for the clarity of its description.

Author Response

Dear reviewer:

Thank you for your recognition of our work and giving me valuable guidance in my manuscript during your busy schedule. We have studied comments carefully and have made answer your comments which we hope meet with approval. In addition, we recorded the details of each modification in the cover letter1. If you still have doubts about our answer, we will revise it again.

Reviewer 2 Report

The manuscript "Identification of three circulating microRNAs in plasma as cinical biomarkers for breast cancer detection" aims to identify and validate hsa-miR-100-5p, hsa-miR-191.5p and hsa-miR.342.3p as biomarkers for breast cancer dectection, suggesting that the combination of the first ones was the optimal combination for BC detection.

The manuscript is interesting and brings novelty. There are some questions to be addressed:

- The aim of the manuscript is not clearly stated nor in the abstract, nor in the introduction. Please provide the aim and the objectives of the study. 

- The workflow of the research (Figure 1) should be in materials and methods and not in the introduction.

- In the results - 2.4. Correlation with clinicopathology features, the authors suggest that the plasma of miR-100.5p was elevated in the luminal subtype but not the HER2 and TNBC subtypes. The authors refer to figure 4. However, in the manuscript there are no mention on luminal A or B and the clinical relevance assciated to the distinction.

- The authors do not provide information regarding the number of BC classified as luminal A or B, HER2 and TNBC. It is important to validate the percentages acoording to the literature in order to validate also the correlation that you obtain.

- In the discussion the authors should mention the difficulties regarding the implementation of miR as biomarkers in BC detection and the advantages of using this model comparing with the established biopsy model pra diagnosis. With the results obtained the authors believe the biopsy of the tissue should be replaced with miR detection?

Author Response

Dear reviewer:

Thank you very much for your comments and suggestions. We seriously thought about your questions and supplemented your suggestions. We appreciate for reviewer warm work earnestly, and hope that the correction will meet with approval. In addition, we recorded the details of each modification in the cover letter2. If you still have doubts about our answer, we will revise it again.

Round 2

Reviewer 1 Report

Dear authors,

Changes made to the text made the manuscript much clearer for the reader.
The presentation of the results is clearer and the discussion more focused.